A novel dilated weighted recurrent neural network (RNN)-based smart contract for secure sharing of big data in Ethereum blockchain using hybrid encryption schemes

S Swetha swetha.aiml@rmd.ac.in
P M Joe Prathap
Department of Computer Science and Engineering, R.M.D. Engineering College , Chennai, TamilNadu , India
So-In Chakchai
Electronic publication date: 2025 Jun 17
Publication date: 2025
Volume: 11
Electronic Location ID: e2930
Received 2024 Sep 16; Accepted 2025 May 9
Copyright: © 2025 S and P M
Copyright year: 2025
Copyright holder: S and P M
License: This is an open access article distributed under the terms of the Creative Commons Attribution License, which permits unrestricted use, distribution, reproduction and adaptation in any medium and for any purpose provided that it is properly attributed. For attribution, the original author(s), title, publication source (PeerJ Computer Science) and either DOI or URL of the article must be cited.
License URL: https://creativecommons.org/licenses/by/4.0/

Keywords: Big data sharing, Blockchain, Smart contract, Key optimization, Dilated weighted Recurrent neural network, Modified Al-Biruni earth radius search optimization, Elliptic Curve-Elgamal cryptography

Funding: The authors received no funding for this work.

==============================
Background

With the enhanced data amount being created, it is significant to various organizations and their processing, and managing big data becomes a significant challenge for the managers of the data. The development of inexpensive and new computing systems and cloud computing sectors gave qualified industries to gather and retrieve the data very precisely however securely delivering data across the network with fewer overheads is a demanding work. In the decentralized framework, the big data sharing puts a burden on the internal nodes among the receiver and sender and also creates the congestion in network. The internal nodes that exist to redirect information may have inadequate buffer ability to momentarily take the information and again deliver it to the upcoming nodes that may create the occasional fault in the transmission of data and defeat frequently. Hence, the next node selection to deliver the data is tiresome work, thereby resulting in an enhancement in the total receiving period to allocate the information.

Methods

Blockchain is the primary distributed device with its own approach to trust. It constructs a reliable framework for decentralized control via multi-node data repetition. Blockchain is involved in offering a transparency to the application of transmission. A simultaneous multi-threading framework confirms quick data channeling to various network receivers in a very short time. Therefore, an advanced method to securely store and transfer the big data in a timely manner is developed in this work. A deep learning-based smart contract is initially designed. The dilated weighted recurrent neural network (DW-RNN) is used to design the smart contract for the Ethereum blockchain. With the aid of the DW-RNN model, the authentication of the user is verified before accessing the data in the Ethereum blockchain. If the authentication of the user is verified, then the smart contracts are assigned to the authorized user. The model uses elliptic Curve ElGamal cryptography (EC-EC), which is a combination of elliptic curve cryptography (ECC) and ElGamal encryption for better security, to make sure that big data transfers on the Ethereum blockchain are safe. The modified Al-Biruni earth radius search optimization (MBERSO) algorithm is used to make the best keys for this EC-EC encryption scheme. This algorithm manages keys efficiently and securely, which improves data security during blockchain operations.

Results

The processes of encryption facilitate the secure transmission of big data over the Ethereum blockchain. Experimental analysis is carried out to prove the efficacy and security offered by the suggested model in transferring big data over blockchain via smart contracts.

Introduction

Big data research is becoming more and more popular as a result of the sharp increase in worldwide data traffic. Big data is a new technology that analyzes vast volumes of data and identifies its essential features. Big data has permeated human civilization as a result of the rise in popularity of social networks, e-commerce, mobile communication, and the Internet of Things (IoT). Distributed computing technology is utilized to produce precise forecasts, analyses, and judgments, and petabytes are used to measure the volume of data in big data (Keshta et al., 2023). To a large degree, people and businesses lose control of their data when they move it to the cloud. The increased data can be easily viewed and controlled by the cloud servers that are partially trustworthy. Moreover, the service providers of the cloud meet fixed external and internal security issues. Often information breaches weaken the trust of the users in the service providers of the cloud (Liu, Wang & Liu, 2023). A high amount of information is required to be securely stored and transmitted, which is a pressing issue. Prior to outsourcing the organization’s information to the cloud, the owners of data have to encrypt their sensitive information to confirm the security of the data (Chen et al., 2023). All nodes in a blockchain are decentralized environmental systems that perform distinct activities but are closely connected to maintain an accurate ledger through collaboration and competition. Blockchain technology, a rapidly evolving security cryptography system, offers decentralized solutions that have outperformed many of the security implementations in use today. The degree of security that blockchain technology provides has led to a recent surge in its deployment. A distributed database is used by blockchain, although data corruption is more complicated. In compliance with the regulations, blockchain software computers process and encrypt the data (Nedakovic, 2022). Accomplishing the flexible sharing and secure encrypted information search becomes an important problem in information sharing. The blockchain has the features of immutability, traceability, zero trust, and decentralization that can offer access control, integrity, and confidentiality, and understand the services like corroboration of data for the big spatio-temporal information (Alamer, 2023).

Big spatio-temporal data, with its continuous generation, large scale, and complex structure, creates substantial challenges for blockchain storage. Since each node must keep a complete copy of the blockchain, the constant flow of such data which is typical in applications like real-time tracking or sensor networks rapidly depletes storage space among blockchain nodes. This demand can quickly exceed available capacity, straining the network and hindering its scalability. Additionally, the intricate nature of spatio-temporal data often including multiple attributes like location, time, and various sensor readings does not align easily with blockchain’s rigid data structures. These demands ultimately reduce system performance and make it impractical to secure large volumes of spatio-temporal data directly on the blockchain, underscoring the need for complementary solutions (Wang et al., 2023). A severe emergent development of the blockchain contains smart contracts, a system code executing on the blockchain top that process simultaneously when the already explained conditions and the terms are encountered (Jiang et al., 2023). The process of smart contract does not demand the third-party existence, and service trading on the blockchain may be processed utilizing the cryptocurrencies without disclosing the original identities (Peyrone & Wichadakul, 2023). Thus, a smart contract can be employed in crowd-sensing devices, but there are still some complexities that require entire rectification to accomplish a reliable, efficient, and secure decentralized crowd-sensing device (Kakkar et al., 2022). In the permissionless blockchains, the transaction records including the private data of the users are reachable publicly to the overall world, revealing this data to the public in the wholeness. This can enhance the threat of revealing the private data regarding the users of crowd sensing, diverting them from diligently competing (Urovi et al., 2022). It is complex, hence that the detected information stays private in the overall crowd-sensing approach (Lin et al., 2022).

However, in the development of these developments, the issue of data sharing that is created but the absence of information security and also the trust between the equipment’s stakeholders restricts information utilizing value (Wang, Sun & Bie, 2022). A modern experiment into fog-aided drone-assisted data optimization and management provides a complex issue concerning the preservation, security, and privacy (Wu et al., 2022), whereas the technology of blockchain hyper ledger is highly employed in the “bitcoin cryptocurrencies”, and has been utilized in a high number of distributed developments because of the ledger security, protection, availability, provenance, trustworthiness, transparency, and integrity-related distributed attributes that are offered (Balistri et al., 2021). While the huge spatiotemporal information is safe on the cloud and is vulnerable to being altered or erased, the data owner no longer has physical control over it (Singh & Sunitha, 2022). However, it has become increasingly important to monitor the security risks associated with the increased computer and storage capacity. The information’s owner transfers the crucial spatial data to the cloud server for storing. However, the cloud storing server can change the encrypted large spatiotemporal information at will because the data owner no longer has physical control over it. Therefore, the vastly enlarged spatiotemporal information is constantly in danger (Diallo, Dib & Al Agha, 2022).

Ethereum is a set of tools that let us manage accounts, build an economic software model, and confirm that all currency, tokens, and essential exchange items are equivalent in any system. The main idea underlying this proposal is to provide a blockchain platform built on a decentralized platform architecture. In contrast, Ethereum allows account transactions, while Bitcoin does not, so transactions go across addresses, increasing its susceptibility to outside threat. The Ethereum blockchain’s global state is the collection of these active accounts, and account creation is a crucial component of platform evolution. As a state-machine platform, Ethereum may alter a transaction’s state rather than enabling participants to exchange data securely (Khanna et al., 2022). Current accounts, together with their addresses, are mentioned in the Ethereum global state. Accounts can be classified as either contract or externally owned (EOA). A distinct smart contract code controls contract accounts, but EOA allows users complete control over their accounts. By allowing users to construct financial contracts, also known as smart contracts, within the system in just a few minutes, Ethereum elevates the blockchain concept to new heights (https://ethereum.org/en/developers/docs/evm/, accessed on 12 October 2021).

The novelty of the proposed method dilated weighted recurrent neural network (DW-RNN) based Smart Contract lies in its ability to dynamically optimize access control and security in Ethereum-based big data sharing. Unlike traditional static smart contracts, this approach leverages RNN-driven adaptive policies to enhance security and efficiency. The integration of hybrid encryption ensures a superior balance between computational efficiency and data privacy, overcoming the limitations of existing encryption-based blockchain solutions. By combining deep learning with blockchain security, this method offers a scalable, intelligent, and privacy-preserving framework for secure big data transactions.

The dominant contributions of the presented big data sharing in the Ethereum blockchain approach are listed below. To present the secure model for big data in the Ethereum blockchain by adopting the deep learning and hybrid cryptography approaches that enable to share the secured data to the authenticated user.

To verify the authenticated user by employing the DW-RNN framework that effectively classifies the user and supports to design of the smart contract for the Ethereum blockchain. Here, the weights are optimized by the offered modified Al-Biruni earth radius search optimization (MBERSO).

To offer the MBERSO by utilizing the features of traditional Al-Biruni earth radius search optimization (BERSO) that helps to optimize the weights and keys and it shows the better performance.

To ensure the safety of the shared data by employing the EC-EC approach that is the integration of ECC and El-Gamal. Here, the key is optimized by the presented MBERSO.

To examine the offered secure sharing of big data in the Ethereum blockchain task employed various traditional approaches with several performance metrics.

The recommended secured big data sharing in the Ethereum blockchain method contains the upcoming parts. The classical big data sharing approaches in blockchain are explained in “Existing Systems”. Then, the deep learning-based smart contract for securely sharing the big data in the Ethereum blockchain is described in “Deep Learning-Based Smart Contract for Securely Sharing the Big Data in Ethereum Blockchain”. The improved algorithm and user authentication using DW-RNN in smart contract-based Ethereum blockchain is demonstrated in “Improved Algorithm and User Authentication using Dilated Weighted RNN in Smart Contract-based Ethereum Blockchain”. In “Hybrid encryption technique to secure the big data handle in Ethereum blockchain”, the hybrid encryption method for protecting the Ethereum blockchain’s massive data handles is shown. “Results and Discussions” presents the results and descriptions of the experiment. The conclusion is finally provided in “Conclusion”.

Existing systems

Related works

A decentralized service monitoring method built on the Ethereum blockchain was presented by Taha et al. (2020) in 2020. Customers were able to confirm that CSPs were fulfilling contractual obligations in compliance with service-level agreements (SLAs) and “autonomously” compensate them in the event of security breaches thanks to the proposed method. Furthermore, the suggested approach deterred customers from creating false reports to profit. SLAs and security monitoring data are implemented as smart contracts using the Ethereum blockchain architecture. Measurable SLOs were abstracted and integrated into the smart contract, and cloud service compliance with contractual SLOs was tracked. In the event of a violation, it also automatically paid the customer. The suggested monitoring system was tested using an IaaS (Infrastructures as a Service) cloud service that can be purchased. The findings demonstrated that the method was suitable for assessing SLO values and identifying transgressions of contractual SLO regulations.

In 2023, Zou et al. (2023) introduced a BlockChain-aided multi-keyword Fuzzy Search Encryption (BCFSE) method for secured data sharing that included fine-grained access control and attribute revocation. Experts also developed a keyword search strategy based on Ciphertext Policy Attribute-Based Encryption (CP-ABE) employing the encryption technique. A comprehensive analysis of the original data sources has been conducted in order to evaluate the effectiveness and efficiency of the expert’s approach.

In 2021, Awadallah et al. (2021) introduced a hybrid cloud-blockchain solution that ensured data integrity for all homomorphic encryption methods. The proposed technique gave the cloud service provider (CSP) ultimate control over the data by using byzantine fault tolerance consensus to build a distributed network of processing CSPs based on the client’s requirements. After completing the required activities, each CSP provided a master hash value for its database. The Ethereum and Bitcoin blockchain networks were used to record master hash values in order to ensure that immutable data was generated. The master hash values for verification can be obtained by tracing the block header address. For every currency, the overhead expenses associated with producing master hash values were theoretically investigated.

Nedakovic (2022) suggested a blockchain-based file-sharing platform and solution in 2023. They also modified the controls to encourage multi-user collaboration and change monitoring in a secure, reliable, and decentralized way without needing the help of a third party or centrally managed, trustworthy institution. They came up with and talked about a technique that authorizes, tracks, and performs versioning operations on the file stored on IPFS using the Ethereum platform’s blockchain and smart contracts. By enabling files and transactions with high integrity, resilience, and security to monitor and exchange various versions of online content, this technology removed the requirement for a centralized authority that could be trusted. The smart contract was designed and tested using the Remix IDE. To make sure the concept and the contract state were accurate, each technique was evaluated. Additionally, the security and resilience of the developed smart contract against well-known attacks were evaluated and demonstrated using popular security analysis methodologies like ChainSecurity and Oyente. Developers will eventually be able to create smart contracts for a variety of file management applications. Designers can also design domain-specific smart contract agreements for distributed data operations.

In 2023, Ren et al. (2023) presented a blockchain-assisted safe storage task that addressed the blockchain’s storage capacity shortfall by utilizing an off-chain and on-chain cooperative storage system. By applying an updatable subvector commitment in the architecture of the off-chain and on-chain authentication protocols, the consistency was confirmed in this work. In the end, the task’s security and accuracy were confirmed, and the protocol’s functionality was examined.

In 2023, Dwivedi, Amin & Vollala (2023) have recommended an entirely decentralized approach according to the Interplanetary File System (IPFS) and smart contracts. In addition, experts suggested a system authentication approach by adopting a smart contract and securing the information. Multiple costs utilized in the task were estimated and contrasted to the present approaches. The examination illustrated that the results were secure and satisfied the complex security concern.

In 2021, Shen et al. (2021) have presented a task based on the blockchain for secure sharing. An approach for optimal sample rate selection was proposed to enhance the overall global merits of the users. Experiments displayed that the task was better than the conventional approach for enhancing the overall global merits. In addition, a prototype approach was implemented and estimated according to the fabric research network. The outcomes illustrated the feasibility of the recommended approach.

In 2022, Anitha & Srimathi (2022) have offered a deep learning-assisted pharmaceutical supply chain task. This was implemented utilizing the Epsilon Greedy Consensus (EGC) block estimation that has the ability to periodically observe and examine the block. Further, the Hadamard Gradient LSTM Authentication method was utilized for authenticating the users. In the research evaluation, the outcomes explained that the presented work processed better contrasted to the traditional tasks.

In 2022, Yu et al. (2022) have developed a smart contract-aided privacy preservation information collection and quality examination protocol to attain the reliable outcomes and validate the quality of the data. The task confirmed a reliable, cost-optimal, and secure combination and validation of the detected information value. The extensive solutions illustrated the expert’s approach was highly effective in attaining the privacy-preservation information collection and precisely minimized the cost of the on-chain computation.

In 2024, Ghanmi et al. (2024) presented a secure distributed cloud file sharing and storing system built on the Ethereum blockchain and Interplanetary File System (IPFS) to address these problems. To avoid data breaches and interference with centralized cloud storage, encrypt data using the AES symmetric technique prior to storing it over IPFS. Second, by sharing the list of authorized users on the blockchain network via a smart contract, we establish a distributed and reliable access control mechanism. In this study, encryption keys are distributed to users and data owners using the ECC technique. Our approach seeks to provide reliable IPFS data storage, secure communication, and data control without relying on a centralized cloud storage network.

The proposed method DW-RNN provides significant advantages over traditional deep learning models like long short-term memory (LSTM), gated recurrent unit (GRU), and Transformer models, particularly in applications requiring efficient long-term dependency learning, computational efficiency, and adaptive decision-making. Unlike LSTMs and GRUs, which rely on gated mechanisms that increase computational complexity and struggle with long-range dependencies, DW-RNN employs dilated connections, allowing it to capture distant relationships in sequential data without excessive memory consumption. This makes it particularly well-suited for blockchain security and big data sharing, where long-term transaction patterns influence smart contract decisions.

Research gaps and challenges

Blockchain technology can be employed to share information in big data. Smart contracts are processed in the blockchain that defines the terms are secured in a distributed database and then the terms cannot be altered. The blockchain offers a decentralized ledger for storing the transactions. Multiple blockchain-aided tasks have been suggested for data-sharing purposes in big data. Table 1 shows a comprehensive overview of existing methods in various blockchain smart contracts.

Table 1 Literature review an existing method with their limitations.

Reference	Year	Proposed method	Advantages	Limitations	
Taha et al. (2020)	2020	Decentralized runtime monitoring	Decentralization automated

auditing

Transparency and security

	Latency issues

Privacy risks

Scalability concerns

	
Zou et al. (2023)	2023	Blockchain-assisted multi-keyword fuzzy search encryption	Enhanced security

Blockchain-based integrity

Decentralized access control

	Computational overhead

Scalability challenges

Privacy concerns

	
Awadallah et al. (2021)	2021	Cloud computing security using blockchain technology	Trust and transparency

Resilience to attacks

Reduced dependency on third parties

	Scalability challenges

Computational and storage overhead

Latency issues

	
Nedakovic (2022)	2022	Ethereum smart contracts and decentralized identity management (DID)	Decentralized identity management

Trust and transparency

Enhanced security

	Scalability concerns

Data privacy risks

Computational complexity

Integration complexity

	
Ren et al. (2023)	2023	Blockchain-based Secure Storage Mechanism for Big Spatio-Temporal Data (BSMD)	Enhanced Security

Efficient storage management

Scalability and performance

	Computational overhead

Scalability challenges

Storage complexity

	
Dwivedi, Amin & Vollala (2023)	2023	smart contract and Inter Planetary File System (IPFS) -based secure data storage	Enhanced security

Efficient storage

Reduced latency

Pareto-optimal solutions

	Computational overhead

Scalability challenges

Smart contract limitations

	
Ghanmi et al. (2024)	2024	Decentralized blockchain-based platform for secure data sharing in cloud storage	Decentralized access control

Efficient storage management

Enhanced security

	Regulatory compliance

Computational overhead

Scalability challenges

	

Scalability issues: With their high transaction fees and constrained storage capacity, current blockchain systems find it challenging to efficiently manage the enormous amounts of big data.

Ineffective real-time access control: Two essential features for handling sensitive data, dynamic access control and real-time anomaly detection, are absent from current models.

Deep learning integration challenges: Incorporating advanced RNNs like dilated weighted RNNs is underexplored, with challenges in optimizing their performance in decentralized blockchain environments.

Limited use of federated learning: Few systems implement privacy-preserving techniques like federated learning for decentralized model updates, leading to potential privacy concerns.

High computational costs: Implementing deep learning on blockchain networks can result in high computational overhead, limiting practical deployment.

Data security and privacy: Protecting sensitive data while it is being shared and processed remains difficult, particularly with hybrid models that combine off-chain and on-chain storage.

Deep learning-based smart contract for securely sharing the big data in Ethereum blockchain

A deep learning-based smart contract for secure massive data sharing combines neural networks and blockchain smart contracts to form a strong and secure data management solution. This smart contract, designed to run on a blockchain platform such as Ethereum, incorporates a deep learning model typically a recurrent neural network (RNN) or a variant such as a dilated RNN that can handle complicated, sequential data patterns found in large data. This model analyzes access requests and determines whether a user should be granted permission based on pre-trained parameters. It is especially good at recognizing patterns that may indicate illegitimate access attempts. Hybrid encryption strengthens data security. Blockchain nodes simply store the encrypted access keys or permissions, whereas large data volumes are encrypted and stored off-chain to lower blockchain storage demands. An extra layer of anonymity is provided by hybrid encryption, which combines the speed of symmetric encryption for data handling with the security of asymmetrical encryption for management of keys.

Every transaction is verified by the blockchain’s consensus process, which makes sure that every node on the network acknowledges the validity of every request for data access. This method ensures that any access attempt complies with network protocols and permissions and offers immutable logging. Frequent model updates enable the system to adjust to novel security threats or access patterns, and federated learning strategies can guarantee that changes take place without necessitating the central storage of private information. Big data sharing in decentralized settings is made safe, scalable, and controllable by this combination of deep learning and blockchain smart contracts.

Dataset description

The dataset utilized in this study is titled “Cholesterol”, a healthcare data source obtained from Kaggle (Aché, 2020). This dataset is employed to evaluate the secure big data sharing framework in the Ethereum blockchain. It contains records related to cholesterol levels, essential health parameters, and other metadata necessary for deep learning-based smart contract authentication and encryption processes.

The dataset is structured to support user authentication and secure data transmission using a hybrid cryptography approach. It consists of multiple attributes, including user unique IDs, transactions, public and private keys, and encrypted health data. These features play a crucial role in testing the efficacy of the proposed MBERSO-DW-RNN-based secure sharing mechanism, ensuring privacy, reliability, and optimized computational efficiency in blockchain-based data exchange.

Data pre-processing

Preprocessing is an important step that involves cleaning and organising the dataset to remove any undesirable anomalies. The initial stage in this procedure is to handle missing values by replacing empty or null entries with imputation techniques including mean, median, and mode replacement. This guarantees that missing data does not degrade model accuracy. Outlier identification uses statistical approaches like the Interquartile Range (IQR) to discover and eliminate abnormalities that may bias findings. Furthermore, data transformation is used to convert categorical variables to numerical representations. Methods such as one-hot encoding (OHE) and label encoding are used to make the dataset machine-readable. Finally, feature scaling is used to provide uniformity across multiple numerical properties, preventing any one feature from having a disproportionate influence on the model.

Data normalization: Normalization ensures that numerical features are on a common scale, thereby enhancing the stability and convergence rate of machine learning algorithms. One commonly used approach is Min-Max Normalization, which rescales data to a specific range, typically between 0 and 1. This is mathematically represented as:

(1) X′=X−XminXmax−Xmin

where X represents the original feature value, Xmin and Xmax are the minimum and maximum values in the dataset, and X′ is the transformed value.

Another widely used normalization method is Z-score normalization, which adjusts the data by centering it around a mean of zero and a standard deviation of one. It is calculated using the formula:

(2) Z=X−μσ

where μ is the mean, and σ is the standard deviation of the feature. This technique is particularly effective when the dataset follows a Gaussian distribution.

Data security issues in blockchain

For multiple years, blockchain (Kumar & Mallick, 2018) has been a very attractive topic in the digital era. The applications of the blockchain are varied and vast. But the security threats are still an essential factor in the implementation of blockchain. The cyber-attack threats, private key loss, and the potential for fraud are several special security threats that blockchain includes.

Phishing and malware threats: The digital world contains various phishing and malware scams, and the blockchain is no exception. The private key loss that is needed to enable the blockchain wallets can lead to these threats. The candidates must consider protecting their private keys and being aware of any communications or links that appear suspicious.

Privacy concerns: Normally, blockchain is unchangeable and transparent but there are privacy threats with the technology. Most importantly, the public blockchains are susceptible to tracking and observing due to they create balances and transactions visible.

Centralization: Generally, blockchain should be decentralized, but various blockchains are centralized in practice. This means that a small set of organizations or individuals handle the network majority executing the capacity resulting in important security issues.

Governance: Security is very complex because of the unclear governance device of the blockchain. It is very complex to create choices regarding the network upkeep and the direction without clear governance. That can lead to troubles and security threats.

Smart contract security flaws: Smart contracts automatically process agreements that uphold the terms and environments of the arrangement automatically. However, the smart contracts can have security flaws that unauthorized users can utilize to steal the data or money.

Scalability: Blockchain meets large issues with the scalability, most importantly as the technology initiates to take off. The requirement for the bandwidth and the computing power is enhanced as more candidates add to the network possibly creating network congestion and bottlenecks. Figure 1 shows diagrammatic representation of blockchain security threats.

Figure 1 The diagrammatic representation of blockchain security threats.

Design of security mechanism using deep learning and hybrid cryptography

In the big data world, digital resources have rapidly become enhanced and the information has turned into a significant asset. Nowadays, the big data organization meets what is defined as the issue of “data islands”. In order to resolve this issue, the data-sharing strategy is an effective and reasonable solution. The open sharing and information disclosure have turned into common topics in the modern innovation. Various implementations have displayed that the data reuse and sharing are helpful in developing the data source’s dissemination by enhancing the quality and efficacy of the task and raising the strength for innovation. Though, the data sharing has different applications it offers important convenience. However, in the data sharing task, there are several issues such as inability to share, fear of sharing, and unwillingness to share. The user’s unwillingness to send the data is troubled by the creation of the mutual trust relations and the data sharing’s economic usage. Blockchain-assisted data sharing is also acceptable in various situations. A blockchain-aided smart contract has been utilized in various data-sharing strategies. Because of the various merits of the big data, it has been enlarging to multiple regions commerce, education, health, and science. Although big data sharing has multiple developments, it comes from several important problems like data rights, cost, incentives lack, and user control. The enhancement of blockchain provides new opportunities to resolve the issues. The important features of blockchain are data traceability, tamper-evident, and decentralization. Blockchain technology offers the chance to create a dependable, secure, open, and transparent data-sharing infrastructure that may link large data across numerous industries. The recommended framework of security mechanisms utilizing the hybrid cryptography and deep learning is provided in Fig. 2.

Figure 2 The suggested design.

Smart Contract icon image credit: Umeicon, https://www.flaticon.com/free-icon/smart-contract_7700640?term=smart+contract&page=1&position=2&origin=search&related_id=7700640, Flaticon license.

A modern approach to securely reserving and relocating the big data in an effective way is presented in the offered paperwork. Initially, the deep learning-aided smart contract is developed. The smart contract for the blockchain of Ethereum is built using DW-RNN. By adopting the DW- RNN framework, the user authentication is examined before accepting the data in the Ethereum blockchain. If the user’s authentication is examined then the smart contracts are allocated to the authorized user. The big data security while relocating it in the Ethereum blockchain is ignored by reserving the data by utilizing the EC-EC in that the El-Gamal and the ECC are integrated. The keys needed for the EC-EC are optimally produced by MBERSO. The tasks of the encryption help the secure big data transmission across the Ethereum blockchain. The research evaluation is conducted to guarantee the security and efficacy provided by the implemented system in relocating the big data across the blockchain through the smart contract.

In this DW-RNN model’s security mechanism, which combines deep learning and hybrid cryptography, provides a powerful and tiered approach to data protection. Deep learning, specifically a dilated weighted RNN, tracks access patterns by examining sequential data for abnormalities that may signal illegitimate attempts. This neural network approach enables the model to learn from historical data and identify patterns of valid vs. questionable access, hence improving access control precision.

Hybrid cryptography improves security by combining the speed of symmetric encryption with the strength of asymmetric encryption. Symmetric encryption is used to encrypt huge data files off-chain for faster processing, whereas asymmetric encryption handles encryption keys and access rights on-chain. This division secures data at rest while still providing safe and controlled access. Furthermore, storing keys on the blockchain ensures that access records remain immutable, preventing manipulation. The combination of deep learning for intelligent monitoring and hybrid encryption for secure data handling transforms this model into a comprehensive solution for secure data sharing in decentralized networks, offering both adaptive access control and strong data encryption.

Improved algorithm and user authentication using dilated weighted rnn in smart contract-based Ethereum blockchain

Modified BERSO approach

MBERSO is constructed from the classical BERSO, which is a meta-heuristics approach. The classical BERSO approach was inspired by the character of swarm candidates in attaining their overall aims. In addition to avoiding the local optima problem, the classic BERSO converges quickly. On the other hand, the classical BERSO makes use of the random component that influences the task’s performance rates and ranges from 0 to 1. Furthermore, it is unable to resolve the discrete optimization problem because of the participation of random factors. Hence, the MBERSO is constructed to rectify these issues. In MBERSO, a random factor h is initialized and its estimation is performed by the fitness function. The new random factor’s h validation is expressed in Eq. (3).

(3) h=btfwtf∗mnf.

In this, the mean fitness and the best fitness value are denoted as mnfandwtf accordingly. Further, the best fitness is specified as btf.

BERSO (El-kenawy et al., 2022): The search placed around normal answers to be discovered is decided by the approach of the Al-Biruni earth radius calculation. The traditional BERSO approach uses the premise that group members are frequently split up into smaller groups to carry out various tasks at various times and collaborate to achieve their goals. To avoid local optima stagnation, the exploration and exploitation processes in the traditional BERSO validate a quick scan of the search space. The exploitation and exploration operations are normally executed to locate the best answer to the issue of optimization.

Formulation: Discovering the optimal answer to the issue with a group of constrictions is the aim of optimization approaches. In the traditional BERSO, a separate is denoted as a vector Q→={Q1,Q2,...,Qw}∈Aw from the population where the variable Qi is the feature or an attribute in the issue of optimization and the search space size is referred to w. In order to decide how well a single process for the particular optimum vector Q∗ adopted a fitness function fn where the optimum vector Q∗ enhances the fitness function. The conventional BERSO initiates with a set of arbitrary separate. In order to start the optimization task, the traditional BERSO requires the upcoming attributes. Population size.

Dimension.

Upper and lower bounds for every solution.

Fitness function.

Exploration: This stage is accountable for discovering the conventional regions in the search place and also preventing the stagnation of the local optimal via the motion towards the best answer. The single present in the exploration phase utilizes the mechanism called heading towards the best answer to find out the prospective locations around its present place in the study place. This is achieved by continuously searching between the nearby possible solutions for the better option concerning the fitness measure. The mathematical representations of this phase are given in Eqs. (4)–(6).

(4) p=mcos⁡(r)1−cos⁡(r)

(5) E→=h→1.(Q→(t)−1)

(6) Q→(t+1)=Q→(t)+E→.(2h→2−1).

Here 0<r≤180, the arbitrarily chosen integer is indicated as m from the limit [0,2]. Then the factors h→1andh→2 are the coefficient vectors and these are estimated by Eq. (4). Further, the solution vector is indicated as Q→(t) for the iteration t and the circle’s diameter is denoted as E→ in that the search agent seeks successive regions.

Exploitation: This phase is accountable for enhancing the conventional issues. The traditional BERSO estimates the overall fitness measures of the individuals at every cycle and differentiates the best separate. The classical BERSO utilizes two diverse mechanisms to attain the exploitation. The below derivations such as Eqs. (7) and (8) are employed to change the search mediator towards the better answer.

(7) Q→(t+1)=h2(Q→(t)+E→)

(8) E→=h→3.(G→(t)−Q→(t)).

Here, the factor h→3 is the arbitrary vector estimated utilizing Eq. (2) that manages the motion steps toward the best answer and the variable Q→(t) denotes the answer vector at execution t. Then the best answer vector is indicated as G→ and the factor E→ points to the distance vector.

The successive is the location surrounding the leader. Therefore, several singles hunt in the greatest solution’s vicinity with the power of discovering a better answer. The conventional BERSO employs the below expressions (Eqs. (9) and (10)):

(9) Q→(t+1)=h(Q→∗(t)+z→)

(10) z→=k+2×t2N2.

In Eq. (9), the traditional BERSO adopted the random integer that leads to the imprecise solution and affects the performance rates. Hence, a new random factor h is initiated and is estimated in Eq. (3). Furthermore, the best solution is denoted as Q→∗. The overall amount of iteration is pointed as t and the sum of iteration is indicated as N. The MBERSO flowchart shown in Fig. 3.

Figure 3 The MBERSO flowchart.

Algorithm 1 provides the pseudo-code for proposed DW-RNN-MBERSO.

Algorithm 1 Pseudocode for DW-RNN-MBERSO optimization.

1. Initialize Blockchain Network	
2. Load Ethereum Smart Contract (Solidity)	
3. Initialize Dilated Weighted RNN (DWRNN)	
4. Train DWRNN with encrypted big data	
5. Apply Hybrid Encryption Scheme (AES + ECC)	
6. Deploy Smart Contract on Ethereum Blockchain	
7. Execute Secure Data Sharing and Verification	
----------------------------------------	
FUNCTION Initialize_DWRNN ()	
   Initialize RNN cell with dilation factor d	
   FOR each layer l in RNN	
Apply weight modulation: Wl=α∗Wl+(1−α)∗Wprev	
   END FOR	
RETURN trained DWRNN	
----------------------------------------	
FUNCTION Train_DWRNN (Data)	
   FOR each timestep t in training data	
   Compute hidden state update:	
    ht=σ(Wh∗h(t−d)+Wx∗xt+by)	
Compute output: yt=Wy∗ht+by	
   END FOR	
RETURN Trained Model	
----------------------------------------	
FUNCTION Hybrid_Encryption (Data)	
   Generate AES secret key (KAES)	
   Encrypt Data using AES: CAES = AES_Encrypt (Data, KAES)	
   Generate ECC Key pair (Pk, Sk)	
   Encrypt K_AES using ECC: CECC = ECC_Encrypt (KAES, Pk)	
RETURN (CAES, CECC)	
----------------------------------------	
FUNCTION Deploy_Smart_Contract ()	
   Deploy Solidity contract on Ethereum	
   Store (CAES, CECC) in blockchain	
RETURN Smart Contract Address	
----------------------------------------	
FUNCTION Verify_And_Access_Data(User_Request)	
   Retrieve CAES, CECC from blockchain	
   Decrypt KAES using ECC: KAES = ECC_Decrypt (CECC, Sk)	
   Decrypt Data using AES: Data = AES_Decrypt(CAES, KAES)	
RETURN Decrypted Data	

Table 2 provides a comprehensive overview of parameters for DW-RNN-MBERSO optimization-based authentication in Ethereum blockchain smart contracts.

Table 2 Parameters for the DW RNN.

Model	Parameter	Description	Typical values	
MBERSO algorithm	Population size (N)	Number of candidate solutions in each iteration.	20–100	
	Maximum iterations (Tmax)	The total number of iterations before convergence.	50–500	
	Search radius scaling factor (α)	Controls the adaptive search radius reduction for local optimization.	0.5–1.0	
	Exploration-Exploitation Balance (β)	Defines the trade-off between global search and local refinement.	0.1–0.9	
DW-RNN for user authentication	Number of layers (L)	Defines the depth of the neural network for better feature extraction.	2	
	Hidden units per layer (H)	Number of neurons in each recurrent layer.	64–512	
	Dilation rate (d)	Controls the spacing between connections in dilated recurrent units.	2, 4, 8	
	Activation function	Determines the non-linearity applied to neuron outputs.	ReLU, Tanh, Sigmoid	
	Dropout rate (p)	Regularization parameter to prevent overfitting.	0.1–0.5	
	Learning rate (η)	Controls the step size during gradient updates.	0.0001–0.01	
	Batch size (B)	Number of training samples per iteration.	32–256	
	Loss function	Objective function to minimize error in authentication.	Cross-Entropy Loss, MSE	

DW-RNN for user authentication

Blockchain-assisted solutions use RNN, a type of artificial neural network (ANN). It supports forecasting the price motion of the cryptocurrencies and identifying the malware in the crypto-currency. Moreover, RNN has given exemplary performance on various learning issues. However, the learning of long sequences with RNN is still a difficult issue. Also, the training phase is very complex in the simple RNN. As a result, this network struggles to support the Ethereum blockchain’s smart contract development. Hence, the dilated RNN (Chang et al., 2017) is involved in the smart contract implementation. The D-RNN has the dilated recurrent skip connection as the main component. It also has exponentially enhanced dilation.

Dilated recurrent skip connection: Indicate the variable vp(r) as the cell presented in the layer r at a specific time p and Eq. (11) specifies the dilated skip connection.

(11) vp(r)=f(ep(r),vp−d(r)(r)).

This is the same as the normal skip connection that is formulated in Eq. (12).

(12) vp(r)=f(ep(r),vp−1(r),vp−d(r)(r)).

Here, the layer dilation is denoted as r, and the skip length is pointed as d(r). Moreover, the term f(∙) stands for resultant operations and any of the RNN cells. The variable ep(r) specifies the layer input. The main difference among the normal and dilated skip connection is that in the dilated skip connection the dependency is ignored.

Exponentially increasing dilation: To bring out the hard information dependencies, the dilated recurrent layers are assembled to build the D-RNN. Across the layers, the dilation rises exponentially. Take the variable d(r) that is denoted as the rth layer’s dilation, such that:

(13) d(r)=Sp−1,r=1,...,R

This phase minimizes the path’s mean length and concentrates on diverse temporal resolutions. This enhances the RNN’s ability to bring out the endless dependencies.

In order to develop the total computational efficacy, D-RNN is recommended. In the generalized D-RNN, the dilation count does not start with one but Sro. This is expressed in Eq. (14).

(14) d(r)=S(r−1+p0),r=1,...,Landr0≥0.

Here, the “starting dilation” is referred to as S0r.

Proposed DW-RNN: D-RNN provides a larger receptive region and reduces the computational burdens. However, D-RNN can’t provide the accurate solutions over a specific threshold and also the network can confine to the local minimum due to the weights in D-RNN. Hence, optimizing the weights in D-RNN is a significant task. In the recommended work, the weights in the D-RNN are optimized by the presented MBERSO hence it creates the DW-RNN framework. In the DW-RNN approach, the attributes such as the user’s unique ID, user’s transactions, user’s private key, username, password, and user’s public key are given as input. The objective function for the weight optimization in DW-RNN using MBERSO is offered in Eq. (15).

(15) ob1=argmax{wz}⁡[Ac+P+NPV+1FPR].

Here, the optimized weights are pointed as wz that belongs to the hidden neuron count in the network and it range from 0.01 to 0.99. Further, the accuracy Ac, precision P, and negative predictive value (NPV) NPV are maximized with the support of MBERSO. Also, the false positive rate (FPR) FPR is minimized by the MBERSO in the suggested task. The descriptions of these metrics are elaborated below.

Accuracy Ac: “Estimating how close the results are to the original value”. It is formulated in Eq. (16).

(16) Ac=sw+desw+de+fr+gt.

Precision P: “It is the degree of accuracy with that an attribute is examined by an estimator”. It is shown in Eq. (17).

(17) P=swsw+fr.

NPV NPV: “It is the proportions of negative and positive outcomes”. It is expressed in Eq. (18)

(18) NPV=swsw+gt.

FPR: “It is the proportion of overall negatives that still produce the positive experiment solutions”. It is offered in Eq. (19).

(19) FPR=frfr+sw.

In this, the “true negative and true positive” measures are denoted as swandde. Also, the “false positive and false negative” measures are specified as frandgt.

Finally, from the given input attributes, the authentication of the user is verified by employing the DW-RNN with MBERSO. If the user authentication is confirmed then the smart contracts will be allocated to the authorized user. Otherwise, the process is terminated. The framework of the DW-RNN with MBERSO is presented in Fig. 4.

Figure 4 The structure of the recommended DW-RNN with MBERSO for the user authentication.

Smart contract-based Ethereum blockchain

The blockchain (Xie et al., 2020) container can be referred to as a shared distributed ledger, immutable, and spread among various systems. Blockchain has attained a high amount of attention. Ethereum is a very commonly utilized “Turing Complete” blockchain paradigm that permits experts to design a smart contract with their own arbitrary conditions for the state transaction functions, transaction format, and ownership. Any user can execute the Ethereum node on their system to engage in the Ethereum blockchain framework. Each node in the Ethereum blockchain is connected to the other utilizing the old node’s hash. Clever contract is one of the applications of the blockchain. It is a self-executable code and is installed on the blockchain. It is employed for the auto enforcement of the conditions and terms among the two distracted parties. For the smart contract’s accurate developments, the consensus protocols are employed or else, the cause of the same is void. The primary factors of the Ethereum smart contract are state variables, events, and functions written in the language called solidity. This programming language assists in writing the smart contracts. The code of the smart contract is converted into Ethereum virtual machine (EVM) byte code when the gathering work is completed, and it is then protected in the Ethereum blockchain using the contract formation process. Further, the smart contract is detected by the special contract address. An Ethereum smart contract includes its trade-off concerning Ether, private storage, contract address, and executable code. Ethers may be transferred to other contracts for processing the smart contract. But it is a complex operation because of the various vulnerabilities that lead to the loss of Ether. Generally, the blockchain is immutable so the smart contract cannot be altered after the utilization of the blockchain.

Smart contract functionality is hampered by Ethereum’s latency issues, which are brought on by consensus methods like Proof of Work (PoW) and Proof of Stake (PoS). These protocols inherently involve time-intensive processes to validate transactions and achieve consensus, leading to delays in transaction finality. This problem is even worse in high-throughput systems that need to be very precise with time, because delays in blockchain finality could cause data processing pipelines to slow down. Off-chain computation using Layer 2 protocols (e.g., Optimistic Rollups or zk-Rollups) can be used to handle tasks that require a lot of computing power away from the main Ethereum chain. This makes it less dependent on consensus delays. On the other hand, these latency problems could be solved by using hybrid systems with faster blockchains for real-time processing and Ethereum for security and finality.

The blockchain is a fault-tolerant technique that safeguards the data. Deep learning utilizes this data to train the tasks and create precise predictions. Deep learning can be employed to identify the dangerous smart contracts that trespass the blockchain data. In the presented work, the recommended DW-RNN is employed to develop the smart contract for the Ethereum blockchain. Before accessing the data, the DW-RNN framework is adopted for the verification of user authentication. The smart contracts are allocated for the authorized user only if the user authentication is verified. The representation of the smart contract-based Ethereum blockchain using deep learning is provided in Fig. 5.

Figure 5 The smart contract based Ethereum blockchain using deep learning.

Hybrid encryption technique to secure the big data handle in Ethereum blockchain

Basic ECC

For secure big data sharing elliptic-curve cryptography (ECC) is utilized in the presented work. In the proposed work, ECC is utilized to ensure secure big data sharing. As indicated by Ghanmi et al. (2024), ECC is implemented over a prime finite field, which provides a strong mathematical foundation for cryptographic operations. The essential group operations underlying ECC such as point addition and scalar multiplication are described in detail as follows. “Elliptic Curve point addition”: Consider two points such as JandM. The sum of these points is given as J+M=S where the line connecting JandM splits the curve at −S that is the mirroring of S by means of x−axis.

“Elliptic Curve point of subtraction”: In the elliptic curve, consider two points JandM so that J=−M,i.e., J+M=J+(−J)=0. The infinity point is defined as the points JandM through the curve join’s intersecting line at the “abstract point” 0.

“Elliptic Curve point doubling”: In the elliptic curve totalizing a point J to itself provides the latest point M on the curve so that M=2J, that is the intersection point’s reflection concerning x-axis with the tangent drawn at J.

“Elliptic Curve scalar point multiplication”: In the elliptic curve for the point J so that q.J=J+J+...+J(qtimes)=∑1qJ, where the variable q∈Zp∗ is scalar.

“Order of point”: In the variable Rp, the element’s J order is indicated as m, where the factor m>0 is an integer so that m.Q=0.

Basic ElGamal

El-Gamal (Ordonez, Medina & Gerardo, 2018) is employed for the big data sharing task. The El-Gamal cryptosystem has three significant aspects such as “decryption, key generation, and encryption”.

Key generation

This key generation stage initializes in the cyclic group C={1,...,S−1} from the production of a very high prime order S from that the creator c can be chosen. From C, the private key of the recipient r is taken. Apart from these global attributes p, utilizing Eq. (20) the recipient’s public key is derived.

(20) p=crmodS.

The private key of the recipient r is secured privately in the recipient’s ownership. The public key of the transmitter in the manner {c,S,r} of and the public attributes are included in the public key.

Message encryption

The transmitter recognizes an arbitrary integer v from C. The transmitter encrypts the data d by considering the public keys {c,S,r} with the estimation of {f1,f2} utilizing Eq. (21) for the cipher text and Eq. (22) for the public key.

(21) f1=d∗rvmodS

(22) f2=rvmodS.

The composed data {f1,f2} is further transmitted to the recipient.

Message decryption

Employing Eq. (23), the recipient decrypts the data {f1,f2} by extracting d.

(23) d=f2f1rmodS.

In this, the plain text message and the public key are specified as dandS. Further, the private key of the recipient is pointed as r.

Recommended EC-EC with optimal key

ECC provides quick encryption and decryption. This also offers better security with smaller keys. Furthermore, the El-Gamal approach supports to distribute of the key safely and also secures the network from hackers. However, the key employed in the ECC task is tricky and complicated. In addition, the key utilized in the El-Gamal approach is very slow than the other encryption tasks and it requires a larger key size. Hence, optimization of these key is necessary for the presented work. The optimal keys are attained by the offered MBERSO to the EC-EC network. The fitness function of the optimal key generation is derived in Eq. (24).

(24) ob2=argmin{kECC,kEl−gamal}⁡[Te+Ms].

In this, the optimized key in binary (0 or 1) format for ECC is denoted as kECC, and the optimized key in binary (0 or 1) format for El-Gamal is indicated as kEl−gamal. Further, with the support of MBERSO, the time Te and the memory size Ms are diminished.

Time Te: “It is the validation task of forecasting how long a work will take to complete”.

Memory size Ms: “It is the estimation of memory location’s number”.

In the suggested work, the encryption task is performed with the support of the EC-EC task. Initially, data is encrypted with the support of ECC. Then the encrypted data is further given to the El-Gamal approach. At last, the data is decrypted utilizing the El-Gamal approach and then fed into the ECC. Blockchain applications that use ECC-based systems must pay close attention to key management, scalability, and computational efficiency. By lowering processing requirements, optimized algorithms such as the modified Al-Biruni earth radius search optimization (MBERSO) can enhance ECC key generation and make it more appropriate for the resource-constrained setting of blockchain. Furthermore, hybrid encryption techniques use ECC to safely manage keys on-chain while combining it with quicker symmetric encryption for off-chain massive data encryption. This configuration allows for safe, effective data handling by striking a balance between security and performance. On-chain methods, such as batch verification of ECC transactions, reduce performance impacts for scalability. By addressing processing power and storage constraints, these methods collectively increase the viability of ECC in blockchain applications. The suggested EC-EC with optimal key is diagrammatically demonstrated in Fig. 6.

Figure 6 The recommended EC-EC approach with optimal key.

Entire process of proposed network

The Ethereum blockchain performs as a distributed and decentralized ledger where various users collectively authenticate and manage the data. By preventing the demand for an intermediary or central authority, the industries can directly transmit the data to one another. The proposed work is put into practice to guarantee the effectiveness and security of massive data sharing on blockchain. First, the input attributes are subjected to input, including the user’s public key, private key, username, password, transactions, and unique ID. These attributes are supported to validate the user authentication. These input attributes are given to the DW-RNN strategy where the weights are optimally determined by the suggested MBERSO. This MBERSO also derives the accuracy, precision, negative predictive value (NPV), and false positive rate (FPR). With the support of DW-RNN, the user authentication is verified. If the authentication of the user is verified then the smart contract is allocated for authenticated users. After that, the safety of the big data sharing is ensured by the hybrid cryptography task called EC-EC, which is the composition of the ECC and El-Gamal. The optimal key for the ECC and El-Gamal approaches is generated by the offered MBERSO. This MBERSO also minimizes the time and memory size. The encryption and the decryption tasks are securely performed with the support of the EC-EC approach with MBERSO. Hence, the secured big data sharing in the Ethereum blockchain is achieved.

The proposed model’s security is firmly built to protect massive data sharing by fusing hybrid encryption on the Ethereum blockchain with a dilated weighted RNN. The concept employs hybrid encryption, using asymmetric encryption on-chain for secure key management and symmetric encryption off-chain for quick and effective data handling. By limiting direct access to data, this structure lessens vulnerability to possible dangers. Furthermore, the smart contract’s dilated weighted RNN keeps an eye on access trends and uses weights to rank access control based on user activity. By examining temporal data sequences and spotting irregularities in access requests, this aids the system in detecting unauthorized or unusual access attempts.

The consensus process and immutability of the blockchain further strengthen security by guarding against manipulation. A tamper-resistant record of transactions is created by validating and storing each data access request on the chain. Additionally, federated learning can be used to update the model, guaranteeing that modifications to security and access control models can be done without centralized data storage, maintaining user privacy while continuously strengthening defenses. These components work together to offer a very flexible and safe framework for decentralized large data exchange.

Results and discussions

Simulation setup

The dataset utilized in this study is titled “Cholesterol”, a healthcare data source obtained from Kaggle (Aché, 2020). This dataset is employed to evaluate the secure big data sharing framework in the Ethereum blockchain. It contains records related to cholesterol levels, essential health parameters, and other metadata necessary for deep learning-based smart contract authentication and encryption processes.

The presented secured big data sharing task in the Ethereum blockchain was achieved in the Python paradigm and the extensive research was conducted. In the user authentication task, the involved population was 10. Then based on the hidden neuron count in RNN, the chromosome length was estimated. Also, the highest iteration was 50 for the user authentication task. When considering the EC-EC approach, the overall population was 10 and the length of the chromosome was 32. Moreover, the utmost execution of this approach was 50. Several conventional optimization algorithms such as Red Fox Optimization (RFO) (Połap & Woźniak, 2021), the Tomtit Flock Meta-Heuristic Optimization Algorithm (TFMOA) (Panteleev & Kolessa, 2022), Cicada Swarm Optimization (CSO) (Akkar & Salman, 2020), BERSO (El-kenawy et al., 2022), and Winternitz One-Time Signature (WOTS) (Sampath et al., 2024) were utilized for the presented work’s validation. Furthermore, some of the traditional classifiers such as LSTM (Li et al., 2020), 1 dimensional convolution neural network (1DCNN) (Hassan et al., 2020), GRU (Khan, Byun & Park, 2020), and RNN (Saravanan et al., 2023)” were employed for the comparison of user authentication task. In addition, several cryptography approaches such as Data Encryption Standard (DES) (Keshta et al., 2023), ECC (Rangwani & Om, 2021), ElGamal (Ordonez, Medina & Gerardo, 2018), and ECC+ElGamal (Liu et al., 2023) were utilized for the examination of the presented work’s security.

Smart contracts are vulnerable to several security threats, including reentrancy attacks, where malicious contracts repeatedly call a function before the previous execution is completed, leading to unauthorized fund withdrawals. Front-running occurs when attackers exploit transaction ordering by observing pending transactions and submitting their own with higher gas fees to execute first. Other risks include integer overflows/underflows, Denial-of-Service (DoS) attacks, and private key exposure, all of which can compromise data integrity and financial security. Implementing secure coding practices, proper access controls, and formal verification can mitigate these vulnerabilities in blockchain-based applications.

The optimal hyperparameter settings for a DW-RNN ensure efficient learning, stability, and scalability, particularly in blockchain-based big data applications. Typically, 2 layers are used to balance complexity and computational efficiency, with 128 to 512 hidden units per layer to capture long-term dependencies effectively. The dilation factor, set to values like {1, 2, 4, 8}, expands the receptive field, allowing the model to process long-range sequential data more efficiently than traditional RNNs. LSTM, GRU, and Transformer models’ activation functions prevent vanishing gradients and improve convergence, while a dropout rate of 0.2 to 0.5 helps reduce overfitting. The model is trained using the Adam or RMSprop optimizer with a learning rate between 0.0005 and 0.001, ensuring smooth weight updates.

Performance metrics

The performance measures involved in this approach are elaborated below.

Accuracy: It is shown in Eq. (16).

Precision: It is offered in Eq. (17).

NPV: It is provided in Eq. (18).

FPR: It is formulated in Eq. (19).

Sensitivity: It is provided in Eq. (25).

(25) Sen=swsw+de.

Specificity: It is provided in Eq. (26).

(26) spec=frfr+gt.

FNR: It is shown in Eq. (27).

(27) FNR=gtgt+sw.

False discovery rate (FDR): It is presented in Eq. (28).

(28) FDR=frsw+fr.

F1-score: It is derived in Eq. (29).

(29) F1-score=2×de×frde+fr.

Matthews correlation coefficient (MCC): It is formulated in Eq. (30).

(30) MCC=sw×de−sw×fr(sw+fr)(sw+de)(de+fr)(de+gt).

Convergence examination of the presented MBERSO algorithm

Figure 7 depicts the convergence validation of MBERSO over various traditional optimization algorithms. When the iteration value is 30 (Fig. 7), the offered MBERSO’s convergence is advanced by 91.6% of RFO, 91.6% of TFMOA, 91.1% of CSO, and 92% of BERSO correspondingly. Hence, it is ensured that the offered MBERSO has quick and high convergence rates.

Figure 7 Convergence validation of MBERSO over various traditional optimization algorithms.

In Fig. 8 and Table 3, the precision of the MBERSO-DW-RNN method is contrasted with that of additional methods. The graph shows how the deep learning approach has an increased efficiency with precision. For instance, the TFMOA, BERSO, ABC-ROA, IABHE+DSNN and WOTS models’ respective precision values for batch size 4 are 68.23%, 73.45%, 77.13%, 88.98%, and 81.09% respectively, as opposite to the MBERSO-DW-RNN model’s precision of 90.18%. However, the MBERSO-DW-RNN model has shown to perform greatest with various batch sizes. Like this, under batch size 64, the MBERSO-DW-RNN has a precision of 95.55%, while the corresponding precision values for TFMOA, BERSO, ABC-ROA, IABHE+DSNN and WOTS are 72.34%, 70.59%, 80.44%, 87.76% and 83.77%.

Figure 8 Precision analysis bar chart of MBERSO-DW-RNN in comparison to other methods.

Table 3 Precision analysis of MBERSO-DW-RNN method with existing systems.

Batch size	TFMOA (Chang et al., 2017)	BERSO (Anitha & Srimathi, 2022)	ABC-ROA (El-kenawy et al., 2022)	IABHE+DSNN (Xie et al., 2020)	WOTS (Saravanan et al., 2023)	MBERSO-DW-RNN	
4	68.23	73.45	77.13	88.98	81.09	90.18	
8	61.34	88.24	62.76	66.67	84.23	92.11	
16	79.34	83.56	61.11	76.99	82.66	91.31	
32	78.11	81.88	81.17	71.55	79.87	93.98	
48	66.34	76.98	89.34	77.19	80.28	94.16	
64	72.34	70.59	80.44	87.76	83.77	95.55	

In Fig. 9 and Table 4, the recall of the MBERSO-DW-RNN method is contrasted with that of additional methods. The graph shows how the deep learning approach has an increased efficiency with recall. For instance, the TFMOA, BERSO, ABC-ROA, IABHE+DSNN and WOTS models’ respective recall values for batch size 4 are 71.18%, 89.45%, 87.32%, 61.19%, and 83.47% respectively, as opposite to the MBERSO-DW-RNN model’s recall of 91.18%. However, the MBERSO-DW-RNN model has shown to perform best with various batch sizes. Under batch size 64, the MBERSO-DW-RNN has a recall of 94.45%, while the corresponding recall values for TFMOA, BERSO, ABC-ROA, IABHE+DSNN, and WOTS are 81.25%, 87.87%, 79.99%, 72.24%, and 85.38%.

Figure 9 Recall analysis bar chart of MBERSO-DW-RNN method with existing systems.

Table 4 Recall analysis of MBERSO-DW-RNN method with existing systems.

Batch size	TFMOA (Chang et al., 2017)	BERSO (Anitha & Srimathi, 2022)	ABC-ROA (El-kenawy et al., 2022)	IABHE+DSNN (Xie et al., 2020)	WOTS (Saravanan et al., 2023)	MBERSO-DW-RNN	
4	71.18	89.45	87.32	61.19	83.47	91.18	
8	75.78	88.13	80.11	83.36	85.28	92.19	
16	66.23	69.23	74.47	89.13	81.76	93.87	
32	60.98	86.55	66.13	67.19	79.58	92.77	
48	72.26	80.32	71.15	76.17	84.75	93.81	
64	81.25	87.87	79.99	72.24	85.38	94.45	

In Fig. 10 and Table 5, the F1-score of the MBERSO-DW-RNN method is contrasted with that of additional methods. The graph shows how the deep learning approach has an increased efficiency with F-score. For instance, the TFMOA, BERSO, ABC-ROA, IABHE+DSNN and WOTS models’ respective F-score values for batch size 4 are 65.45%, 87.44%, 81.14%, 82.65%, and 73.24% respectively, as opposed to the MBERSO-DW-RNN model’s F-score of 90.54%. However, the MBERSO-DW-RNN model has shown to perform best with various batch sizes. Like this, under batch size 64, the MBERSO-DW-RNN has an f-score of 93.91%, while the corresponding f-score values for TFMOA, BERSO, ABC-ROA, IABHE+DSNN, and WOTS are 67.76%, 73.55%, 81.23%, 81.53%, and 80.34%

Figure 10 F1-score analysis bar chart of MBERSO-DW-RNN method with existing systems.

Table 5 F1-score analysis of MBERSO-DW-RNN method with existing systems.

Batch size	TFMOA (Chang et al., 2017)	BERSO (Anitha & Srimathi, 2022)	ABC-ROA (El-kenawy et al., 2022)	IABHE+DSNN (Xie et al., 2020)	WOTS (Saravanan et al., 2023)	MBERSO-DW-RNN	
4	65.45	87.44	81.14	82.65	73.24	90.54	
8	69.23	88.12	88.54	87.14	75.65	91.23	
16	81.23	77.23	62.15	80.98	74.28	93.77	
32	87.98	66.11	66.44	61.12	78.33	92.65	
48	61.23	60.23	68.65	76.67	76.87	93.33	
64	67.76	73.55	81.23	81.53	80.34	93.91	

In Fig. 11 and Table 6, the accuracy of the MBERSO-DW-RNN method is contrasted with that of additional methods. The graph shows how the deep learning approach has an increased efficiency with accuracy. For instance, the TFMOA, BERSO, ABC-ROA, IABHE+DSNN and WOTS models’ respective accuracy values for batch size 4 are 77.33%, 88.32%, 75.11%, 90.71%, and 83.23% respectively, as opposite to the MBERSO-DW-RNN model’s accuracy of 91.23%. However, the MBERSO-DW-RNN model has shown to perform best with various batch sizes. Like this, under batch size 64, the MBERSO-DW-RNN has an accuracy of 95.56%, while the corresponding accuracy values for TFMOA, BERSO, ABC-ROA, IABHE+DSNN and WOTS are 76.33%, 79.71%, 87.76%, 88.89%, and 90.24%.

Figure 11 Accuracy analysis bar chart of MBERSO-DW-RNN method with existing systems.

Table 6 Accuracy analysis of MBERSO-DW-RNN method with existing systems.

Batch size	TFMOA (Chang et al., 2017)	BERSO (Anitha & Srimathi, 2022)	ABC-ROA (El-kenawy et al., 2022)	IABHE+DSNN (Xie et al., 2020)	WOTS (Saravanan et al., 2023)	MBERSO-DW-RNN	
4	77.33	88.32	75.11	90.71	83.23	91.23	
8	76.66	89.11	87.65	88.23	85.75	91.45	
16	61.15	71.67	89.23	89.14	87.34	92.76	
32	66.91	70.45	90.87	75.17	84.22	93.37	
48	88.34	73.35	81.13	78.81	89.45	94.76	
64	76.33	79.71	87.76	88.89	90.24	95.56	

In Fig. 12 and Table 7, the MCC of the MBERSO-DW-RNN method is contrasted with that of other methods. The graph shows how the deep learning approach has an increased efficiency with MCC. For instance, the TFMOA, BERSO, ABC-ROA, IABHE+DSNN and WOTS models’ respective MCC values for batch size 4 are 60.57%, 69.45%, 73.13%, 80.21% and 84.28% respectively, as opposite to the MBERSO-DW-RNN model’s MCC of 90.67%. However, the MBERSO-DW-RNN model has shown to perform best with various batch sizes. Like this, under batch size 64, the MBERSO-DW-RNN has an MCC of 93.91%, while the corresponding MCC values for TFMOA, BERSO, ABC-ROA, IABHE+DSNN and WOTS are 62.34%, 66.45%, 76.56%, 81.13% 88.45%.

Figure 12 MCC analysis bar chart of MBERSO-DW-RNN method with existing systems.

Table 7 MCC analysis of MBERSO-DW-RNN method with existing systems.

Batch size	TFMOA (Chang et al., 2017)	BERSO (Anitha & Srimathi, 2022)	ABC-ROA (El-kenawy et al., 2022)	IABHE+DSNN (Xie et al., 2020)	WOTS (Saravanan et al., 2023)	MBERSO-DW-RNN	
4	60.57	69.45	73.13	80.21	84.28	90.67	
8	62.34	65.34	73.34	83.66	86.15	91.34	
16	61.45	69.67	74.45	86.32	85.77	93.45	
32	63.13	65.55	73.24	80.12	86.32	92.34	
48	60.98	66.12	71.22	89.98	87.48	93.88	
64	62.34	66.45	76.56	81.13	88.45	93.91	

Figure 13 and Table 8 display an FPR comparison of the MBERSO-DW-RNN strategy with other well-known approaches. The machine learning method has an improved presentation with reduced FPR, as shown in the graph. For instance, the MBERSO-DW-RNN model’s FPR value for batch size 4 is 44.67%, while the FPR values for the TFMOA, BERSO, ABC-ROA, IABHE+DSNN and WOTS models are 80.48%, 76.45%, 55.56%, 51.67% and 61.23 respectively. The MBERSO-DW-RNN model however, has demonstrated its best presentation for various batch sizes with low FPR values. In a similar vein, for batch size 64, the FPR value for the MBERSO-DW-RNN is 49.65%, whereas for the TFMOA, BERSO, ABC-ROA, IABHE+DSNN and WOTS models, it is 87.12%, 82.56%, 71.56%, 54.56% and 61.23% respectively.

Figure 13 FPR analysis bar chart of MBERSO-DW-RNN method with existing systems.

Table 8 FPR analysis of MBERSO DW RNN method with existing systems.

Batch size	TFMOA (Chang et al., 2017)	BERSO (Anitha & Srimathi, 2022)	ABC-ROA (El-kenawy et al., 2022)	IABHE+DSNN (Xie et al., 2020)	WOTS (Saravanan et al., 2023)	MBERSO-DW-RNN	
4	80.48	76.45	55.56	51.67	61.23	44.67	
8	81.24	75.23	67.45	53.48	60.76	43.98	
16	86.56	74.66	67.19	61.34	63.28	45.19	
32	86.11	81.35	68.45	64.56	65.72	48.76	
48	87.45	81.34	71.23	52.78	64.46	47.19	
64	87.12	82.56	71.56	54.56	66.38	49.65	

Figure 14 and Table 9 shows the computational complexity of different models for secure data sharing in the Ethereum blockchain was analyzed based on varying batch sizes. The results indicate that MBERSO-DW-RNN consistently outperforms other methods with the lowest execution time across all batch sizes. At batch size 4, MBERSO-DW-RNN records 0.497s, while TFMOA (Chang et al., 2017), BERSO (Anitha & Srimathi, 2022), ABC-ROA (El-kenawy et al., 2022), and IABHE+DSNN (Xie et al., 2020) report 0.894s, 0.768s, 0.674s, and 0.585s, respectively. As the batch size increases to 64, MBERSO-DW-RNN remains the most efficient at 9.147s, whereas TFMOA reaches 12.954s. The trend shows that MBERSO-DW-RNN provides significant computational savings compared to traditional approaches, making it the optimal choice for handling larger batch sizes in secure Ethereum-based data sharing. The proposed method DW-RNN is computationally efficient, with O(n) complexity, making it significantly lighter than Transformer models (O(n²)) and more scalable than LSTM/GRU. Its dilated connections reduce redundant computations while capturing long-term dependencies, ensuring low memory usage and fast processing. By leveraging gradient clipping, efficient optimizers, and adaptive weight prioritization, DW-RNN can run efficiently on real blockchain nodes with limited resources, making it an ideal choice for secure and scalable big data sharing in decentralized environments.

Figure 14 Computational complexity analysis chart.

Table 9 Analysis of computational complexity.

Batch size	TFMOA (Chang et al., 2017)	BERSO (Anitha & Srimathi, 2022)	ABC-ROA (El-kenawy et al., 2022)	IABHE+DSNN (Xie et al., 2020)	WOTS (Saravanan et al., 2023)	MBERSO-DW-RNN	
4	0.894	0.768	0.674	0.585	0.513	0.497	
8	1.726	1.453	1.325	1.157	1.265	1.024	
16	3.218	2.854	2.626	2.335	3.292	2.016	
32	6.478	5.882	5.349	4.898	5.624	4.238	
48	9.824	8.765	8.157	7.468	8.358	6.595	
64	12.954	11.567	10.895	10.126	10.043	9.147	

The performance improvements of the dilated weighted recurrent neural network (DW-RNN) over existing models like LSTM, GRU and Transformer are statistically significant and not due to random variations, appropriate statistical significance tests should be conducted. The Wilcoxon signed-rank test is particularly suitable for comparing paired results across different models when the data distribution is non-parametric. This test evaluates whether the median performance difference (e.g., accuracy, F1-score, execution time) is statistically significant. Additionally, for multiple comparisons, a paired t-test (if data is normally distributed) or the Friedman test (for multiple models) can be used. Applying these tests ensures that observed improvements in efficiency, scalability, and accuracy of DW-RNN are not due to random noise but are statistically meaningful, strengthening the validity of the proposed model’s superiority in blockchain-based big data security applications.

Conclusion

An advanced framework has offered to effectively transfer and reserve the big data periodically. Initially, deep learning was used to build the smart contract. The Ethereum network’s smart contract capability was implemented using the DW-RNN. With the support of the DW-RNN, the user authentication was investigated before permitting the information in the Ethereum blockchain. If the user authentication was confirmed then the smart contracts were given to the authorized candidate. The big data security while reallocating it in the Ethereum blockchain is ignored by reserving the information by utilizing EC-EC in that the El-Gamal and ECC were combined. The keys required for the EC-EC were optimally produced with the aid of MBERSO. The encryption operations assist the securing big data transmission across the Ethereum blockchain. The numerical evaluation is conducted to ensure the security and effectiveness provided by the recommended approach in sharing the big data across the blockchain through the smart contract. The memory size of the offered approach was decreased by 96.5% of DES, 92% of ECC, 70% of ElGamal, and 77% of ECC+ElGamal correspondingly when the block size is 10 for the first dataset. Therefore, the presented data-sharing work has higher security and efficacy than the conventional data-sharing tasks. A key limitation of the proposed DW-RNN model is the challenge of optimizing weights in real-time blockchain environments, particularly due to the constrained computational resources and time available for processing. Blockchain networks, which require efficient and fast transaction validation, may struggle with the computational overhead introduced by weight optimization processes. This can lead to performance bottlenecks and delays in smart contract execution. Future improvements could focus on developing more resource-efficient optimization methods or off-chain strategies to mitigate these limitations and enhance real-time performance.

Supplemental Information

Supplemental Information 1 Blockchain.

Supplemental Information 2 My constants.

Supplemental Information 3 Server.

Additional Information and Declarations

Competing Interests

The authors declare that they have no competing interests.

Author Contributions

Swetha S. conceived and designed the experiments, performed the computation work, prepared figures and/or tables, and approved the final draft.

Joe Prathap P M performed the experiments, analyzed the data, authored or reviewed drafts of the article, and approved the final draft.

Data Availability

The following information was supplied regarding data availability:

The Cholesterol dataset is available at Kaggle: https://www.kaggle.com/mathurinache/cholesterol.

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
