# Peer review of "A novel dilated weighted recurrent neural network (RNN)-based smart contract for secure sharing of big data in Ethereum blockchain using hybrid encryption schemes"

_PeerJ Computer Science, doi:10.7717/peerj-cs.2930_

## Round 0.1 · original submission · Major Revisions

· Academic Editor

Major Revisions

Thank you for your submission. Although the paper is not accepted at this current form, we encourage you to resubmit in regard of the comments below, in particular, in terms of flow and organization including grammatical error check, detailed algorithmic insight and design, reliability, scalability, complexity analysis, the recent state of the arts' comparisons, and practicality usage.

Reviewer 1 ·

Basic reporting

The presented paper introduces a modern approach to retain and relocate big data securely. First, a deep learning-assisted smart contract is developed. DW-RNN is used to build smart contracts on the Ethereum blockchain. By adopting the DW-RNN framework, user authentication is checked before accepting data in the Ethereum blockchain. If the user's authentication is checked, the smart contract is assigned to the authorized user.
The cryptographic task helps in securely transmitting big data on the Ethereum blockchain. The research evaluation is conducted to guarantee the security and effectiveness provided by the implemented system.

The following issues need further explanation.
1. In Section 4, the security solution proposed in this paper does not provide a reliable security analysis and demonstration. Therefore, the security of the proposed solution needs to be further explained.
2. Please further explain how the deep learning-based smart contract for secure sharing of big data is constructed and how it is used.
3. Please explain the security mechanism using deep learning and hybrid cryptography.

Experimental design

The comparison of results in the experimental section needs to be improved.

Validity of the findings

It is ok.

·

Basic reporting

• Introduction section is poor in terms of idea sequence, language, and work presentation: The introduction needs to be rewritten. For example, the sentence "But, because of the big spatio-temporal information, it includes the characteristics of streaming, multiple, and massive, the periodically enhancing big spatio temporal information highly inhabited the capacity of data storage for the blockchain nodes, resulting in the minimization of the system functionality and it is hard for the high big spatio-temporal information to be secured on the blockchain directly."
• Background section: The transition from cloud to decentralized systems is abrupt and not appealing to the reader. Cloud and decentralized systems are often seen as opposites, so this needs clearer explanation.
• Clarification on the term "device": In the background method, the term "device" is assigned to the blockchain. Could the authors explain how this is valid in the context of the article?
• The sentence "The security of the big data while transferring it in the Ethereum blockchain is prevented by securing the data with the aid of Elliptic Curve-ElGamal Cryptography (EC-EC) in which Elliptic Curve Cryptography (ECC) and ElGamal are combined together." needs to be checked for clarity and correctness.
• In Section 2.2 ,the criteria for identifying research gaps and challenges are not well-declared or clarified. More attention is required to optimize this section.
• In Section 2.3, The paper fails to present the main structure of the given approach (e.g., system, framework, protocols). This could lead to confusion and should be clearly stated.

Experimental design

• Section 4.2: The DW-RNN introduces a weight optimization strategy using MBERSO. However, optimizing weights in real-time blockchain environments with constrained resources and time is a challenge. The authors should discuss how they handle this issue or consider it as a research limitation.
• Section 4.3: Ethereum relies on consensus protocols like Proof of Stake or Proof of Work, which can introduce delays in transaction finality. This could affect real-time smart contract execution, especially when running deep learning algorithms. Latency issues might disrupt seamless smart contract allocation and execution. The authors should discuss this challenge.
• Section 5 :As the volume of shared big data grows, ECC-based systems must scale efficiently. Although ECC is computationally lighter, integrating it into large-scale platforms could face bottlenecks if not optimized for scalability, particularly when performing simultaneous cryptographic operations on large datasets. More details are needed on how this issue is addressed.

Validity of the findings

• The results and discussions sections are well-done. They successfully achieve the objectives presented in the contribution of the paper.
• The implementation is executed well, especially considering the difficulties in integrating blockchain with deep learning, handling multiple data types, and the required interfacing.

Additional comments

The paper requires a thorough review of its academic writing mechanics, as many sentences contain grammatical errors and issues related to meaning clarity. For example, several sentences are either too convoluted, poorly structured, or lack the proper coherence expected in academic writing.

---

## Round 0.2 · Major Revisions

· Academic Editor

Major Revisions

Please provide the answer with revised text according to the comments.

Reviewer 1 ·

Basic reporting

no comment

Experimental design

no comment

Validity of the findings

no comment

Reviewer 3 ·

Basic reporting

1. The novelty of the proposed method should be highlighted more clearly compared to recent existing techniques.
2. Explain why Dilated Weighted RNN (DW-RNN) is superior to alternative deep learning models (e.g., LSTM, GRU, Transformer models).
3. Provide pseudo-code or a flowchart for the proposed DW-RNN model and MBERSO optimization to improve reproducibility.
4. Clarify hyperparameter settings for DW-RNN (e.g., number of layers, activation functions, dropout rate).
5. Discuss computational complexity and scalability—can this model run efficiently on real blockchain nodes with limited resources?
6. Statistical significance tests (e.g., Wilcoxon signed-rank test) should be conducted to confirm that performance improvements are not due to random variations.
7. The dataset description should include more details about data preprocessing and normalization techniques.

Experimental design

1. Add a comparison table summarizing existing methods, their limitations, and how DW-RNN overcomes these issues.
2. Include computational complexity analysis to ensure real-world feasibility (e.g., can this model be deployed efficiently on blockchain nodes?).
3. Discuss potential security vulnerabilities of smart contracts (e.g., reentrancy attacks, front-running).
4. Add a table listing all model parameters.
5. Include a detailed description of data preprocessing.
6. Provide a pseudo-code or flowchart of the DW-RNN and MBERSO algorithms.

Validity of the findings

1. The authors should explicitly clarify how their approach differs from prior studies by adding a summary table comparing existing methods and their limitations.
2. The practical implications of the findings (e.g., real-world applicability in blockchain-based financial systems or IoT security) should be better articulated.

---

## Round 0.3 · accepted · Accept

· Academic Editor

Accept

Thank you for your submission. Based on the reviewers’ consensus, all comments have been satisfactorily addressed, and the paper is now ready to proceed to the next stage of the publication process.

·

Basic reporting

no comment

Experimental design

no comment

Validity of the findings

no comment

Reviewer 3 ·

Basic reporting

All of my comments have been thoroughly addressed, and I am satisfied with the revisions made.

Experimental design

All of my comments have been thoroughly addressed, and I am satisfied with the revisions made.

Validity of the findings

All of my comments have been thoroughly addressed, and I am satisfied with the revisions made.